# Modelling biological nitrogen fixation in global natural terrestrial ecosystems

Tong Yu[1] and Qianlai Zhuang[1,2]

[1]Earth, Atmospheric, and Planetary Sciences, Purdue University, West Lafayette IN 47907, USA

5  [2]Department of Agronomy, Purdue University, West Lafayette, IN 47907, USA

*Correspondence to: Qianlai Zhuang (qzhuang@purdue.edu)*

**Abstract.** Biological nitrogen fixation plays an important role in the global nitrogen cycle. However, the fixation rate has been usually measured or estimated at a particular observational site. To quantify the fixation amount at the global scale, process-based models are needed. This study develops a biological nitrogen fixation model to quantitatively estimate nitrogen fixation rate by plants in natural environment. The revised nitrogen module better simulates the nitrogen cycle in comparison with our previous model that has not considered the fixation effects. The new model estimates that tropical forests have the highest fixation rate among all ecosystem types, which decreases from the equator to the polar region. The estimated nitrogen fixation in global terrestrial ecosystems is 61.5 Tg N yr$^-$$^1$ with a range of 19.8 - 107.9 Tg N yr$^{-1}$ in the 1990s. Our estimates are relatively low compared to some early estimates using empirical approaches, but comparable to more recent estimates that involve more detailed processes in their modelling. Furthermore, the contribution of nitrogen made by biological nitrogen fixation depends on ecosystem type and climatic conditions. This study highlights that there are relatively large effects of biological nitrogen fixation on ecosystem nitrogen cycling and the large uncertainty of the estimation calls for more comprehensive understanding of biological nitrogen fixation. More direct observational data for different ecosystems to improve future quantification of fixation and its impacts.

## 1. Introduction

In most terrestrial ecosystems, nitrogen (N) available for plants is generally limited although it is the most abundant element in the atmosphere (LeBauer and Tresder, 2008). Nitrogen usually enters terrestrial ecosystems through processes of nitrogen deposition and from biological N fixation (BNF). Nitrogen deposition is a physical process, representing the direct input of reactive nitrogen including organic N, ammonia, and nitrogen oxides ($NO_y$) including nitric oxide (NO), nitrogen dioxide ($NO_2$), nitric acid ($HNO_3$) and organic nitrates from the atmosphere to biosphere. BNF, a biochemical process that converts nonreactive nitrogen ($N_2$) to reactive nitrogen, provides a liaison between the atmosphere and biological systems. Lightning is also a way to convert $N_2$, adding 3-5 Tg N yr$^{-1}$ to terrestrial ecosystems (Levy and Moxim, 1996). Nitrogen input via rock weathering is another important source for terrestrial ecosystems, adding 3-10 kg N ha$^{-1}$ yr$^{-1}$(Morford et al., 2011; Houlton et al., 2018). BNF is significantly greater than lightening induced N fixation (Galloway et al., 1995). On a global scale, anthropogenic nitrogen to the environment could be more than 160 Tg N yr-1 (Gruber and Galloway, 2008), which is even greater than terrestrial N fixation (~110 Tg N yr$^{-1}$). However, taken together, natural N fixation is the primary source in the absence of human activities to global terrestrial ecosystems. For natural terrestrial ecosystems, the amount of N added is approximately balanced by the nitrogen converted back to the atmosphere (Stedman and Shetter, 1983) and lost into ocean and other aquatic systems.

Once entering terrestrial ecosystems, N can be taken up by plants and microbes, and converted into other oxidized forms through mineralization, nitrification and denitrification. In terrestrial ecosystems, $N_2$ fixation generally affects the nitrogen cycle and nutrient level to constrain plant productivity. Any change of nitrogen input to terrestrial ecosystems will influence their soil nitrogen content.

In the process of BNF, $N_2$ is converted to ammonia by certain soil microorganisms which can then be utilized by and incorporated into plants. In natural environment, $N_2$ fixation is conducted by two types of microorganisms:

asymbiotic organisms and symbiotic organisms. The former includes blue-green algae, lichens and free-living soil bacteria (Belnap, 2002; Granhall & Lid-Torsvik, 1975), and the later includes fungi and nodule forming Rhizobium species. Among them, the most dominant fixers are leguminous plants and their N fixation mechanisms are also the best known (Sullivan et al., 2014; Vitousek et al., 2013). A symbiotic relationship exists between legume plants and bacteria, in which legume plants provide the bacteria energy through photosynthesis and the bacteria around the rhizobia-supplies the legume N in the form of ammonia. To date, the amount of N fixation by legumes is estimated in the range of 11.3-33.9 kg N ha$^{-1}$ yr$^{-1}$ (2.8~8.4 g m$^{-2}$ yr$^{-1}$) in natural terrestrial ecosystems.

The biological N$_2$ fixation rate has been usually measured or estimated at a particular observational site. To quantify the fixation amount at the global scale, process-based models and sufficient observational data are needed. This study develops a BNF model considering the symbiotic relationship between legume plants and bacteria. The model is extensively calibrated with site-leveled observational data. The model is then extrapolated to the global terrestrial ecosystems to quantify the fixation rate in the 1990s. The factors influencing the fixation rate are also analyzed for different terrestrial ecosystems, including the distribution of legume plants, soil temperature, and soil properties and types.

## 2. Methods

### 2.1 Overview

We first develop a BNF model and then couple the model with an earlier version of biogeochemistry model quantifying soil carbon and nitrogen dynamics (Yu and Zhuang, 2019). The revised model is then used to quantify the BNF at regional and global scales in natural terrestrial ecosystems. The BNF rate estimates consider the effects of environmental conditions including temperature, soil moisture, soil mineral nitrogen content and soil carbon content. The modified model is calibrated and evaluated with observed N$_2$ fixation rate data from published studies for various natural terrestrial ecosystems from the Arctic to tropical ecosystems. The model sensitivity to model input is analyzed. The model is then extrapolated to the global terrestrial ecosystems at a monthly step and a spatial resolution of 0.5° by 0.5° for the final decade of the 20$^{th}$ century. The effects of physical conditions on BNF are then analyzed.

### 2.2 Model description

The Terrestrial Ecosystem Model (TEM) is a process-based model that simulates carbon and nitrogen dynamics, hydrological and thermal processes for terrestrial ecosystems. Although many efforts were made to incorporate more details of the N cycle, the N input from the atmosphere to ecosystems has not fully been incorporated to date, especially the BNF as input. Here we improve the N dynamics within TEM by considering N$_2$ fixation by legumes. The model schematic and other calculations including carbon cycle and nitrogen cycle are inherited from an earlier version of TEM (Zhuang et al., 2003; Yu and Zhuang, 2019).

BNF is the most significant process in either symbiotic or non-symbiotic forms converting stable molecular $N_2$ into N chemical compounds that are available to plants. For most terrestrial ecosystems, $N_2$ fixers could be in many forms, such as free-living bacteria, lichens, and blue algae. But among them, symbiotic BNF is a dominant process to provide biologically accessible N, and most systematical BNF is regulated by legume plants, especially in

croplands and semi-natural environments (Mus et al., 2016). In natural environments, contributions from legumes can be significant but with large uncertainties, which is greatly determined by various environmental conditions (Lindemann and Glover, 1996). In this study, the $N_2$ fixation via legume plants is modeled considering (1) the accessible N concentration in soils, (2) the limitation of temperature, (3) soil water status, (4) the carbon demand for $N_2$ fixation, and (5) the percentage of $N_2$ fixing plants for each ecosystem type as:

$$N_{fix} = N_{fixpot} f_t f_W f_N f_c f_{plant} \tag{1}$$

where $N_{fix}$ is the nitrogen fixation rate, $N_{fixpot}$ is the potential $N_2$ fixation rate (g N day$^{-1}$) , $f_t$ is the influence function of soil temperature, $f_W$ is the soil water function, $f_N$ is the function of root substrate N concentration, $f_C$ is the function of plant carbon availability, and $f_{plant}$ is the function of legume plant coverage. Please refer to Table 4 for value range of related parameters.

The potential $N_2$ fixation is highly related to the total N demand of plants and the available nitrogen in soils. Theoretically, the definition of potential $N_2$ fixation rate should be the difference between the demand and supply of N. Both of them vary with plant types, stages of growth and soil conditions. For large spatial-scale simulations for various ecosystem types, it is impossible to derive potential $N_2$ fixation because of data availability. $N_{fixpot}$ can be estimated based on dry matter of root, nodule or plant dry matter (Voisin et al, 2003, 2007).

However, root biomass is also difficult to measure directly. In most published studies, the potential nitrogen fixation rate was measured using an acetylene reduction array (ARA) method (Hardy et al, 1968, 1973), and some used $^{15}$N methods (Shearer and Kohl, 1986). In our simulation, $N_{fixpot}$ is assumed to be a constant for each ecosystem type. The $N_{fixpot}$ range is determined from literature and specific values for various ecosystem types are obtained through model parameterization.

Soil temperature is a controlling factor for both microbial activities and plant growth. A large number of studies show that different plants have slightly different preferences for temperature (Montanez et al, 1995; Breitbarth et al., 2007; Gundale et al., 2012). For soybean, 20-35 °C is optimal (Boote et al., 2008), and for white clover the optimal temperature can be 13-26 °C (Wu and McGechan, 1999). The activity of microbes responds slightly differently to temperature among species. For most of them, the optimum temperature is 20-25 °C, and at

12-35 °C the activity is not limited. Generally, the relation between the factor and temperature is not exactly a Gaussian distribution. BNF increases as the temperature rises from minimum temperature (0-5 °C) for N fixation to optimal temperature, maximum rate occurs within an optimal range (15-25°C), and decreases from optimal to maximum temperature above which BNF will stop at 35-40 °C:

$$f_t = \begin{cases} 0 & when \ (t < t_{min} \ or \ t > t_{max}) \\ \frac{t - t_{min}}{t_{optL} - t_{min}} & when \ (t_{min} \leq t < t_{optL}) \\ 1 & when \ (t_{optL} \leq t \leq t_{optH}) \\ \frac{t_{max} - t}{t_{max} - t_{optH}} & when \ (t_{optH} < t \leq t_{max}) \end{cases} \tag{2}$$


where the upper limit ($t_{max}$) is set to 45 °C. There is no lower limit, but when t is low enough, $f_t$ will be close to zero (Wu and McGechan, 1999; Boote et al., 2008; Holzworth et al., 2014) (Table 1). For the convenience in computing, a lower limit is set in our model. When the temperature goes beyond its upper or lower limit, $f_t$ is assumed to be 0.

Water stress has a direct effect on nitrogen fixing system (Sprent, 1972). With proper temperature, soil

moisture condition is the major factor controlling nitrogen fixation rate (Srivastava and Ambasht, 1994). Soil water deficit and flood dramatically inhibits $N_2$ fixation because of drought stress and oxygen deficit, respectively (Omari et al., 2004; Mario et al., 2007). In our model, the water factor is linearly related with soil water content (Williams, 1990; Wu and McGachan, 1999):

$$f_w = \begin{cases} 0 & when \ (W_f \leq W_a) \\ \varphi_1 + \varphi_2 & when \ (W_a < W_f < W_b) \\ 1 & when \ (W_f \geq W_b) \end{cases} \tag{3}$$

where $W_f$ (J kg$^{-1}$) is the available soil water, which is defined as the ratio of water content to that at the field capacity. In soils, water potential generally includes osmotic and matrix potentials, ranging from -0.1 to -0.3 bar for typical soils, which has little effects on the N fixation. But when the soil gets very dry, the potential can be up to - 100 to -200 bar and increases rapidly. $W_a$ is the bottom threshold below which $N_2$ fixation is totally restricted by soil moisture. $W_b$ is the upper threshold above which nitrogen fixation is not limited by soil moisture. $\varphi_1$ and $\varphi_2$ are

parameters representing the linear relationship between soil water content and its effect on $N_2$ fixation, respectively (Table 1).

It is generally thought that more substrate N in soils will slow down the $N_2$ fixation, because plants can uptake N directly from soil with less energy (Vitousek and Field, 1999). By comparison, $N_2$ fixation needs more energy and consumes more carbon than plant N uptake does. Thus, the $N_2$ fixation is only considered to occur when

the direct N uptake from soil cannot meet the plant N demand. In our model, the inhibition effect of N is defined as (Wu and McGehan, 1999):

$$f_N = \begin{cases} 1 - f_{Nup} ln(1000 - N_s) & when \ (N_s \geq 0.001) \\ 1 & when \ (N_s < 0.001) \end{cases} \tag{4}$$

Where $f_{Nup}$ is a parameter related to legume biological $N_2$ fixation and soil N. $N_S$ is the soil mineral N (g N m$^{-2}$). BNF efficiency shows a natural logarithmic relation with the soil mineral N.

N$_2$ fixers get photosynthetic carbohydrate support from plants. Because the product of every unit of nitrogen fixed consumes a certain amount of carbon, the lack of carbon supply will inhibit the N$_2$ fixation. The carbon cost for per unit of fixed N$_2$ varies widely depending on environmental conditions and ecosystem types. For example, the consumption of carbon is only 1.54 times of fixed N$_2$ for cowpea (Layzell et al., 1979), and it can be 6.3 to 6.8 times for soybeans (Ryle et al., 1979). It is also related to the life cycle of plants. The carbon effect is modeled following a Michaelis-Menten equation (Boote et al., 1998):

$$f_C = \frac{1}{1 + K_c / C_r} \tag{5}$$

where C$_r$ is the soil carbon content (g C m$^{-2}$) to represent carbon availability from plants to N$_2$ fixers. K$_c$ is the Michaelis-Menten constant, which is plant species dependent.

## 2.3 Data

The classification of land cover and leguminous biomes were derived from the combination of the International Geosphere and Biosphere (IGP) land-cover classification system and the study of Schrire et al (2005). The experimental N$_2$ fixation data for model calibration were collected for 7 major ecosystem types. Nitrogen fixation rates were determined with acetylene reduction assay (ARA) method in most published studies (Table 2, data were from Cleveland et al. (1999)), expressed in kg N m$^{-2}$ yr$^{-1}$. Some of them were measured with the $^{15}$N natural abundance technique.

The parameters for N$_2$ fixation module were initialized with a priori values (Table 2). Ecosystem-specific and microbe guild-specific parameters were inherited from previous TEM model (Zhuang et al., 2003; Yu and Zhuang, 2019). The global simulations were conducted at a spatial resolution of 0.5 by 0.5 degree and at a monthly time step. Historical climate data including temperature, precipitation, cloudiness and water vapor pressure were derived from the Climate Research Unit (CRU) (Mitchell and Jones, 2005). Soil texture data were from Melillo et al. (1993) and Zhuang et al. (2003). Other initial conditions including vegetation properties, soil carbon content and soil nitrogen contents were from Chen and Zhuang (2013) and Zhuang et al. (2012).

For regional simulations, the total amount of fixed N$_2$ was also influenced by legume coverage. For each ecosystem type, we estimated the coverage according to the distribution of legume plants and field studies (Table 3, the coverage data are compiled from Cleveland et al. (1999)), where the minimum and maximum values were derived from the abundance of N$_2$-fixers.

## 2.4 Model calibration and site-level validation

Most model parameters are legume-specific or vegetation-specific and are adjusted based on value ranges from previous studies (Table 1). Model is parameterized for 7 representative natural terrestrial ecosystems (Table 2). Root mean square error (RMSE) and coefficient of determination ($0 \leq R^2 \leq 1$) were used for model calibration. RMSE was calculated to show the mean difference between simulated data and observational values. The model is iterated with

changing parameters until the RMSE reached a certain value for each site. Most parameters in the model driving nitrogen cycle in the soil have been defined and calibrated in previous studies (Yu and Zhuang, 2019). The calibrated model is evaluated at the site level and then extrapolated to the global terrestrial ecosystems.

## 2.5 Model sensitivity and uncertainty Analysis

The response of $N_2$ fixation of different biomes to input data and variation of parameters was analyzed using sensitivity testing. Four major input variables were selected, including air temperature, precipitation, soil nitrogen content and soil organic carbon content. The monthly average input variables were changed by ±10% of the original level for each site and each grid. The variables were changed at 6 levels, respectively, and the rest of input variables were kept at their original values. The sensitivity was calculated by comparing the simulated annual nitrogen

fixation to the simulations with the original input values.

## 3. Results

### 3.1 Model evaluation

To evaluate the model, thirty-five observational sites were selected for 7 major ecosystem types across the

globe, representing different climate and soil conditions. The experimental data of $N_2$ fixation have a mean value of 12.9 kg N ha$^{-1}$ yr$^{-1}$, with a standard deviation of 17.7 kg N ha$^{-1}$ yr$^{-1}$. The maximum observed fixation occurred in temperate forest in New Zealand, while the minimum rate was also for temperate forest in the state of Idaho in the US. Our simulations are comparable with the observed data for all major ecosystem types with the coefficient of determination ($R^2$) of 0.44 and with a slope of 0.46 (Figure 2). The regression results are mainly influenced by some

observed data greater than 30 kg N ha$^{-1}$ yr$^{-1}$. By removing the outliers of observational data, the slope of regression increases to 0.72. Observational data for temperate forests show the greatest variation among all major ecosystem types, with a maximum value reaching 800 times of the minimum one. Simulations are closer to the observations across sites in temperate forests with $R^2$ of 0.26 and slope of 0.42. Our model underestimated nitrogen fixation rate in temperate forests. The large variation in observations may be due to the distribution of legume plants, different

sampling time periods (e.g., growing and non-growing seasons), and varying climate conditions. For tropical forests, our model estimates of $N_2$ fixation are higher than observations with the slope of 0.75 and $R^2$ of 0.44.

### 3.2 Model sensitivity analysis

The model sensitivity analysis quantifies the impact of changes in forcing data on nitrogen fixation rate. Climate conditions including air temperature and precipitation, and soil characteristics of nitrogen content and

carbon content varied at 3 levels to examine the sensitivity. The response of nitrogen fixation rate emissions is quantified for each ecosystem type. The sensitivity test was conducted for all observational sites (Table 2). Temperature is the most sensitive variable (Figure 1). Nitrogen fixation is more sensitive to the change of all forcing conditions. Increasing soil nitrogen results in a lower $N_2$ fixation. Abundant soil nitrogen content inhibits BNF activity, but stimulates nitrification and denitrification processes.

**3.3   Biological nitrogen fixation in global terrestrial ecosystems**

       Tropical forests in South America, Central Africa and South Asia show a wide range of $N_2$ fixation rates between 1 and 200 kg N ha$^{-1}$ yr$^{-1}$ (Bruijnzeel et al, 1991). Here all plants in tropical rainforest are assumed to fix nitrogen and one set of parameters are applied for all tropical forests. The coverage for tropical forests in the landscape was assumed to be 15% (Cleveland et al., 1999), ranging from 5% to 25%. The $N_2$ fixation rate was
estimated to be 18.2 kg N ha$^{-1}$ yr$^{-1}$, which is the highest among all vegetation types. Our simulations show that the total fixed nitrogen ranges from 10.8 Tg N yr$^{-1}$ to 54 Tg N yr$^{-1}$, with the average value of 32.5 Tg N yr$^{-1}$(Table 3). Nitrogen fixation in tropical forests is almost half of the global total amount and a principal contributor of BNF in natural ecosystems. Tropical forests have the largest potential to fix nitrogen given that the optimal temperature and soil moisture for BNF is relatively easy to have under tropical climatic conditions.
Temperate forests cover the largest land area from 30°N to 60°N, including temperate coniferous forest, temperate deciduous forest and temperate evergreen forest. Temperate areas have the majority of legumes and many temperate ecosystems are considered to be N limited. Comparing to other ecosystem types in temperate regions, conifers are likely to limit the reproduction of legumes (Wheatley et al, 2010). In general, plant species carrying nitrogen fixers are only distributed in a small percentage of natural temperate forests, like clear-felled areas and
pastures (Boring and Swank, 1984). Cleveland et al. (1999) indicated that the legume coverage ranges from 1% to 10% of the land area only. Consequently, our simulations indicate that $N_2$ fixation by temperate forests was 12.7 kg N ha$^{-1}$ yr$^{-1}$. The estimates of the total nitrogen fixation were between 1.9 and 19.14 Tg N yr$^{-1}$ (Table 3). Nitrogen fixation in temperate areas contributes 12.5% of the global total amount.

       Savanna covers over a half of African continent, Australia and large areas of South America. It is an important
biome in the Southern Hemisphere. There is a great variation in native legume species. Only in humid savanna, legumes may significantly contribute to the increase of soil nitrogen (Cech et al., 2008). On average, 15% of the vegetation in savanna is considered as legume grass and biological nitrogen fixation occurs when precipitation is greater than 10 mm per month. Generally, nitrogen fixation in savanna is restricted by soil moisture, while temperate grassland is limited by both temperature and soil moisture (Bustamante et al, 1970). Nitrogen fixers are not abundant
for these biomes (Woodmansee et al., 1981). The coverage of nitrogen fixers was assumed to be from 5% to 25%, (Cleveland et al., 1999).  Our simulation assumed that nitrogen fixers cover 15% of the land, resulting in 1.9 kg N ha$^{-1}$ yr$^{-1}$ fixation, representing a much smaller fraction compared to forest ecosystems. Total fixed nitrogen in grasslands appeared to range from 0.62 to 3.1 Tg N yr$^{-1}$, with an average of 1.86 Tg N yr$^{-1}$. For savanna, the total

contribution was less due to its relatively small area. The minimum, average and maximum values were estimated to be 0.45, 1.34 and 2.23 Tg N yr$^{-1}$, respectively.

In tundra and boreal forest regions, both host plants and their rhizobia are adapted to the environment with low temperature. Nitrogen fixation rate is extremely variable for boreal ecosystems. For tundra, the coverage was assumed to be 3-15%, and for boreal forest, the coverage was 4-18%. But in general, the low temperature and permafrost conditions limit the activity of nitrogen fixers (Alexander, 1981). We estimated that tundra ecosystems

fix nitrogen at 3.2 kg N ha$^{-1}$ yr$^{-1}$. Their total BNF was between 0.51 to 2.55 Tg N yr$^{-1}$ with average of 1.54 Tg N yr$^{-1}$. In boreal forests, the fixation rate was much lower (2.1 kg N ha$^{-1}$ yr$^{-1}$) compared to temperate forests.

The fixation could be neglected in deserts because of the extremely dry conditions. Only few legumes may exist in deserts and their growth is highly depended on precipitation events. Even in semi-arid areas, the N$_2$ fixation rate is much lower than that in tropical and temperate forests (5.7 kg N ha-1 yr-1).

Mediterranean ecosystems such as in southern California and some areas in southern Australia are characterized with mild rainy winter and hot dry summer, containing both evergreen and deciduous shrublands, in which nodulated legumes are prominent (Sprent et al., 2017). These legumes are more active in comparatively wet season than in dry season (Sánchez-Diaz, 2001). The ability to fix nitrogen is considered to be one of the most important features that enable legumes and plants to survive under severe environments (Crisp et al., 2004). We estimated that

the N$_2$ fixation rate of these legume species is similar to that in grasslands (2.7 kg N ha$^{-1}$ yr$^{-1}$).

Spatially, the highest rate of N$_2$ fixation occurred in the tropical and sub-tropical areas, as a result of proper climate and soil characteristics for fixers (Figure 3). N fixation from tropical forests and xeric shrubland contributes to nearly half of the global terrestrial amount (Table 3). A lower N$_2$ fixation rate was in high latitudes of eastern China, North America and Europe, which were mainly covered with temperate forests. Compared to tropical areas,

N$_2$ fixation in temperate regions shows a larger variability depending on vegetation types. The spatial variation could be attributed to the distribution of legume plants, in addition to the difference of humidity and temperature conditions. N$_2$ fixation in temperate regions accounts for 35% of the total fixed N$_2$.

Our model estimated that high BNF rates in the growing season is consistent with other regional and global estimates (Cleveland et al., 1999, 2013; Lee and Son, 2005; Lett and Michelsen, 2014). The energetic cost for active

N uptake becomes lowest when soil temperature is around 25℃ (Fisher et al. 2010). Similarly, our estimates of high BNF rates also occur at similar temperature conditions in spring and summer. The global soil nitrogen mineralization rate was estimated to be 696 Tg N yr$^{-1}$ while 15% of plant N demand was provided by BNF (Cleveland et al., 2013). Our estimates of BNF were lower than the estimates by Cleveland et al. (2013) and fell within 10% of the total soil mineralization rate. This result also indicates that about 10% of the mineralized N was

induced by BNF.

During 1990-2000, our simulations show that BNF in natural terrestrial ecosystems is 61.5 Tg N yr$^{-1}$, but anthropogenic N$_2$ fixation was much higher at 140 Tg N yr$^{-1}$ (Galloway et al., 2002). This large amount of anthropogenic N input to terrestrial ecosystems is expected to inhibit the natural BNF and might lead to less BNF in the future.

## 4. Discussion

### 4.1 Comparison with other estimates of biological nitrogen fixation (BNF)

There is a large uncertainty in estimating the N input into terrestrial ecosystems, especially from BNF (Sutton et al., 2014) (Table 3). In our study, a calibrated process-based model was applied to estimate site-level and global BNF in natural terrestrial ecosystems. Empirical models provide reasonable estimation based on relationships between $N_2$ fixation rates and environmental factors (e.g. evapotranspiration) (Cleveland et al, 1999), while process-based approaches consider processes in BNF affected by multiple controlling factors (Fisher et al., 2010; Gerber et al., 2008; Meyerholt et al., 2016). Our estimated BNF in the global terrestrial ecosystems is 61.5 Tg N yr$^{-1}$ with an uncertainty ranging from 19.8 to 107.9 Tg N yr$^{-1}$, which is lower than most existing studies. Cleveland et al. (1999) provided a central value of 195 Tg N yr$^{-1}$ by scaling up field-based experimental data, with a range of 100 - 289 Tg N yr$^{-1}$. This range represents potential distribution of nitrogen fixation. In reality, $N_2$ fixation is also affected by climate and soil conditions, making the actual terrestrial BNF smaller than the potential one. In a more recent study of Cleveland et al. (2013), a total of 127.5 Tg N yr$^{-1}$ was estimated to be related to BNF, based on the relationship between BNF and evapotranspiration (ET). Galloway et al (2002b) also provided several estimates for global BNF. Galloway et al. (2004) further suggested a range of 100 - 290 Tg N yr$^{-1}$ and implied that the true rate of BNF would be at the low end of this range without large-scale human disturbance. In an earlier study (Galloway et al., 2002b), the mean annual global BNF was estimated to be 89-100 Tg N yr$^{-1}$. By assuming a steady state between N input to and loss from ecosystems, Vitousek et al. (2013) estimated the BNF to be 58 Tg N yr$^{-1}$ with a plausible range of 40 - 100 Tg N yr$^{-1}$, which is similar to our estimates. However, Xu-Ri and Prentice (2017) estimated that the $N_2$ fixation was about 340 Tg N yr$^{-1}$ which is almost 5 times larger than our estimates. In their study, BNF was determined by plant N requirement across all biome types.

In our estimation, tropical forests significantly contribute to the total BNF, which is up to 18 kg N ha$^{-1}$yr$^{-1}$. This result is highly related to the density of leguminous plants, and the physical conditions in tropical areas (Crews, 1999). Our simulated results are comparable to the estimates of symbiotic $N_2$ fixation from tropical evergreen (5.5-16 kg N ha$^{-1}$ yr$^{-1}$) and deciduous forests (7.5-30 kg N ha$^{-1}$ yr$^{-1}$) (Reed et al., 2011). Barron et al. (2010) directly measured $N_2$-fixing root nodules across lowland tropical forests and their observations also showed a large variation among individual trees. For a mature forest matrix, the average value was around 10 kg N ha$^{-1}$ yr$^{-1}$, but it could be as high as 200 kg N ha$^{-1}$ yr$^{-1}$ for some areas. Cleveland (2013) provided a similar estimate to ours (around 12 kg N ha$^{-1}$ yr$^{-1}$), but higher values (20-30 kg N ha$^{-1}$ yr$^{-1}$) in their earlier studies (Cleveland et al., 1999). Sullivan et al (2014) analyzed human's impact on tropical N fixation and found, depending on forest ages, fixation was 5.7 kg N ha$^{-1}$ yr$^{-1}$ with a range from 1.2 to 14.4 kg N ha$^{-1}$ yr$^{-1}$, which is lower than our estimates.

For temperate and boreal forests, we estimated that BNF fixation is 2.1-18 kg N ha$^{-1}$ yr$^{-1}$. The existing BNF estimates from literature also show a large uncertainty for those forest ecosystems. For instance, LM3V-N model (Gerber et al., 2009) suggested that the N input to forests to be less than 5 kg N ha$^{-1}$ yr$^{-1}$. But their model also estimated that, in moist forests, the uptake of N could be 30-80 kg N ha$^{-1}$ yr$^{-1}$. Deluca et al. (2002) reported that

cyanobacterium and feather moss could act as a supplement to $N_2$ fixation in boreal forests (0.5 kg N ha$^{-1}$ yr$^{-1}$) while

the organic N accumulation could be 3 kg N ha$^{-1}$ yr$^{-1}$. For the forests in northwest Rocky Mountain, $N_2$ fixation

amount is on average between 0.5 and 2 kg N ha$^{-1}$ yr$^{-1}$ (Clayton and Kennedy, 1985; Fahey et al., 1988) while Kou-

Giesbrecht and Menge's model (2019) estimated the $N_2$ fixation rate to be 0 -10 kg N ha$^{-1}$ yr$^{-1}$ for temperate forests,

and 0 to 6 kg N ha$^{-1}$ yr$^{-1}$ for boreal forests.

There could be a number of reasons for our comparatively lower estimates. The most important one is that there

is a considerable uncertainty in estimating the coverage of $N_2$ fixing plants. High diversity in the distribution of

legume plants highly influences the estimation of total plant coverage, because our estimation was based on site-

level experimental data. In order to improve our understanding, more investigation on legume plant distribution and

associated data for $N_2$ fixers is needed, especially in the Middle Asia, South America and Africa.

Large variations of BNF rates exist across terrestrial ecosystems spatially (Figure 3). The global BNF spatial

pattern is similar to other estimates (Cleveland et al., 1999; Xu-Ri and Prentice, 2017). The highest $N_2$ fixation rate

in tropical regions (more than 50% of the global terrestrial $N_2$ fixation) is primarily due to their warm and moist soil

conditions. Further, $N_2$ fixed by human activities became increasingly influential in the past century (Galloway et

al., 2002), especially in temperate regions due to their large human population.  The anthropogenic N deposition

contributed more to soil N than BNF did. As a result, soils became N rich, inhibiting BNF in temperate soils. This

could explain why the potential $N_2$ fixation rate was high in temperate ecosystems, but only contributed to 20% of

the total fixation.

## 4.2   Major controls on biological nitrogen fixation

    In our simulations, the $N_2$ fixation was primarily influenced by soil temperature, moisture and soil nitrogen

content. The highest $N_2$ fixation rate in tropical ecosystems is consistent with our sensitivity analysis for temperature

and soil moisture. The sensitivity analysis indicated that a 1-3°C increase of temperature led to 7% increase in $N_2$

fixation rate. Nitrogen cycle responds differently between different biomes and legume types. But in general,

increasing temperature will accelerate processes in the N cycle. Soil moisture correlates with BNF in a similar way

with temperature. A slightly increase of precipitation (10%) increased the nitrogenase activity. However, the

response of $N_2$ fixation to soil water stress is not as sensitive as that to the change in temperature.  Xeric shrubland

and savanna in dry tropical areas still contribute greatly to the global $N_2$ fixation, while the contribution of boreal

forests, with low temperature, is much lower.

    BNF is highly regulated by soil nitrogen content. N-deficiency conditions usually favor BNF activities, for

example, in xeric shrubland and savanna. Enhancing soil N content will decrease the $N_2$ fixation rate, which is also

consistent with our sensitivity analysis. It costs less energy for plants to take up N directly from soils rather than

biologically fixing it from the atmosphere (Cannell and Thornley, 2000). However, there is an exception for some

areas in tropical ecosystems. Many tropical soils are comparatively rich in nitrogen, but $N_2$-fixing plants are still

active to compensate the nitrogen depletion due to the rapid N cycling (Pons et al., 2007). This explains why N

fertilization inhibits the BNF in temperate ecosystems, but BNF is still active in N-rich soils in tropical ecosystems.

In areas where the energetic cost succeeds the demand of N, the BNF rate will be comparatively lower. Sullivan et al. (2014) suggested that there were lower rates of BNF in undisturbed mature forests and higher rate in secondary forests, depending on the balance between N-demand and energy consumption.

### 4.3 Model limitation and future work

The incorporation of BNF into TEM allows us to more adequately simulate nitrogen cycle from natural terrestrial ecosystems. However, there are several limitations in this study.

First, the current model ignores the effect of free-living BNF. Although symbiotic BNF is critical for most natural and semi-natural ecosystems, asymbiotic organisms play an important role in extreme environments such as waterlogged soils and deserts. The importance of symbiotic BNF or fixation by leguminous plants may not be as significant as previously thought. Elbert et al. (2012) suggested that cryptogam contributed nearly half BNF in

terrestrial ecosystems, which was up to 49 Tg N $yr^{-1}$. In some tropical areas, the spatial N input from free-living bacteria even exceeds symbiotic input (Sullivan et al., 2014). In addition, legumes are not the only source of symbiotic BNF. Some fungi species have the ability to actively fix atmospheric nitrogen. But in most existing models, fungi or mycorrhizae symbioses are not considered due to the limited knowledge about their mechanisms of fixing N (Fisher et al., 2010). A more comprehensive model that covers various types of nitrogen fixation is needed.

Second, the BNF process in our model is calibrated with a limited amount of data, imposing a general set of parameters to all plant species and soil conditions within an ecosystem type. More observational data from natural terrestrial ecosystems is desirable to improve our model.

Third, it is difficult to isolate the N addition via natural processes from human activities. In the US, 20-35% of annual N input into terrestrial ecosystems are human-related (Sobata et al., 2013). As a result, the quality of

observational data varies from site to site, some BNF data are only semi-natural. The observational data are imperfect, which might have also biased our estimates through model parameterization process.

### 5. Conclusions

This study developed a process-based biological nitrogen fixation model and coupled it with an extant biogeochemistry model. The model was evaluated with observed data for $N_2$ fixation. The model was then

extrapolated to the global natural terrestrial ecosystems. Our model estimates that biological nitrogen fixation in natural terrestrial ecosystems was 61.5 Tg N $yr^{-1}$ during the last decade of the 20[th] century and the greatest fixation rate occurred in tropical regions. Soil temperature, rather than soil moisture and nutrient content, is the most dominant control on $N_2$fixation. Lacking the knowledge about the distribution of $N_2$ fixing plants and their physiological features might have biased our estimates of biological nitrogen fixation at the global scale.


**Data availability**

Climate data including monthly cloudiness, precipitation, temperature, and water vapor pressure are from the Climate Research Unit (CRU) http://www.cru.uea.ac.uk/data (last access: May 2017). Global vegetation data and

soil data are available in Zhuang et al. (2003) and McGuire et al. (2001). The explicit spatial data on soil water pH

from the ORDL gridded soil properties product (https://daac.ornl.gov/cgi-bin/dsviewer.pl?ds_id=546, last access: May, 2020) are based on the World Inventory of Soil Emission Potentials (WISE) database (Batjes, 2000). The global average carbon dioxide concentration is observed at NOAA's Mauna Loa Observatory. N deposit data are from NADP monitor and CASTNET. The initial values of soil microbial carbon and nitrogen, and the ratio of C/ V / N at the global scale, were from a compilation of global soil microbial biomass carbon, nitrogen, and phosphorus

data (https://doi.org/10.3334/ORNLDAAC/1264, last access: May, 2017). *The data presented in this paper can be accessed through our research website (http://www.eaps.purdue.edu/ebdl/).*

**Author contribution**

    Q. Zhuang and T. Yu designed the research. T. Yu performed model simulations and data analysis. Both

authors contributed to the paper writing.

**Competing interests**

    The authors declare that they have no conflict of interest.

**Acknowledgments**

    This study is supported through projects funded by the NASA Land Use and Land Cover Change program (NASA-NNX09AI26G), Department of Energy (DE-FG02-08ER64599), the NSF Division of Information & Intelligent Systems (NSF-1028291). Thanks to Rosen Center for Advanced Computing (RCAC) at Purdue University for computing support.

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

**Table 1.** Description of parameters used in the model

| Parameters | Description | Unit | Reference Value | reference |
|---|---|---|---|---|
| **N_fix** | nitrogen fixation rate | g N m$^{-2}$ day$^{-1}$ | | |
| **N_fixpot** | potential nitrogen fixation rate | g N m$^{-2}$ day$^{-2}$ | 0.01-1×10-3 | Thornley (2001); Eckertsten et al.(2006); Corre-Hellou et al. (2007); Corre-Hellou et al. (2009); |
| **ft** | soil temperature factor | °C | | |
| **t_min** | the minimum temperature for the start of N fixation | °C | 0.5~5 | Boote et al. (2008) |
| **t_max** | the maximum temperature for the stop of N fixation | °C | 40~45 | Boote et al. (2008) |
| **t_optL** | lower threshold of optimal temperature | °C | 10~20 | Boote et al. (2008) |
| **t_optH** | upper threshold of optimal temperature | °C | 25~35 | Boote et al. (2008) |
| **fw** | soil water factor | | | |
| **φ1** | coefficient for soil moisture | | 0 | |
| **φ2** | coefficient for soil moisture | | 2 | APSIM, EPIC (Sharpley and Williams, 1990; Bouniols et al., 1991; Cabelguenne et al., 1999); SOILN (Wu and McGechan, 1999) |
| **Wa** | lower threshold of water content below which N fixation is totally restrict by rhe defict of soil water | | 0 | APSIM, EPIC (Sharpley and Williams, 1990; Bouniols et al., 1991; Cabelguenne et al., 1999); SOILN (Wu and McGechan, 1999) |
| **Wb** | upper threshold of water content above which N fixation is not limited by rhe defict of soil water | | 0.5 | APSIM, EPIC (Sharpley and Williams, 1990; Bouniols et al., 1991; Cabelguenne et al., 1999); SOILN (Wu and McGechan, 1999) |
| **Wf** | available soil water content to that at field capacity | | | |
| **f_Nup** | parameter relating legume biological nitrogen fixation and soil nitrogen content | | 0.01~0.1 | SOILN model (Wu and McGehan, 1999) |
| **Ns** | Soil mineral nitrogen content | g N m$^{-2}$ | | |
| **f_N** | Soil mineral N effect | | | |
| **f_C** | Soil carbon effect | | | |
| **Cr** | Carbon concentration in the soil | g C g$^{-1}$ soil | | |
| **Kc** | Michaelis-Menten Constant for carbon | g C m$^{-2}$ | 0.001~0.01 | Thornley (2001); Eckertsten et al. (2006) |

**Table 2**. Calibration Sites of Biological Nitrogen Fixation Rate for Representative Ecosystems

| Site name | Ecosystem Type | Lon. | Lat. | Experimental method | Reference | N Fixation Rate kg N ha$^{-1}$ yr$^{-1}$ | Simulation, kg N ha$^{-1}$ yr$^{-1}$ |
|---|---|---|---|---|---|---|---|
| Stordalen, Sweden | Tundra | 18 | 68 | ARA[*] | Christie (1987); Sonesson et al. (1980) | 2 | 2.5 |
| Truelove Lowland, Canada | Tundra | -84.5 | 75.5 | ARA | Chapin et al. (1991) | 3 | 2.8 |
| Niwot Ridge, Colorado, US | Tundra | -105.5 | 40 | $^{15}$N | Bowman et al. (1996) | 4.9 | 5.1 |
| Central Sweden | Boreal Forest | 18 | 60 | ARA | Nohrstedt (1985) | 0.93 | 0.9 |
| PNFI, Ontario, Canada | Boreal Forest | -77 | 45.5 | ARA | Hendrickson (1990) | 0.25 | 1.2 |
| Southern British Columbia, Canada | Boreal Forest | -119 | 49 | ARA | Hendrickson and Burgess (1989) | 2.8 | 2.2 |
| Robson moralines, British Columbia | Boreal Forest | -119 | 53 | ARA | Blundon and Dale (1990) | 1.1 | 1.2 |
| Umea, Sweden | Boreal Forest | 19.5 | 64 | ARA | Huss-Danell (1976) | 1 | 1.5 |
| Coweeta Basin, | Temperate Forest | -83 | 35 | N accumulation | Boring and Swank (1984) | 48 | 19.5 |
| Hoh River, Washington, US | Temperate Forest | -123.5 | 48 | ARA | Luken and Fonda (1983) | 40 | 13 |
| Tom Swamp, Massachusetts, US | Temperate Forest | -75 | 42.5 | unspecified | Schwinzer (1983) | 35 | 25.7 |
| Big Creek Basin, Melbourne, Austrilia | Temperate Forest | 145.5 | 38 | ARA | Adams and Attiwill (1984) | 24 | 23.2 |
| Jebo Creek, Utah, US | Temperate Forest | -112 | 42 | $^{15}$N | Skujins et al. (1987) | 10.2 | 12.5 |
| Karri Forest, south-western Austrilia | Temperate Forest | 116 | -34.5 | ARA | Grove and Malajczuk (1992) | 7.93 | 8.5 |
| Woodhill Forest, New Zealand | Temperate Forest | 174.5 | -37 | N accumulation | Baker et al. (1986) | 80 | 23.5 |
| Gainesville, Florida, US | Temperate Forest | -82 | 30 | N accumulation | Permar and Fisher (1983) | 10.6 | 12.8 |
| Fox park, Wyoming, US | Temperate Forest | -106 | 41 | ARA | Fahey et al. (1985) | 13 | 12.5 |
| Mount Robson, Canada | Temperate Forest | -119 | 53.1 | ARA | Blurdon and Dale (1990) | 1.65 | 3.2 |
| Dwellingup, South-western Austrilia | Temperate Forest | 116 | 33 | ARA | O'Connel and Grove (1987) | 2.5 | 3.1 |
| Adair, Oregon, US | Temperate Forest | -123 | 44.6 | ARA | Heath et al (1988) | 0.74 | 2.4 |

| Site | Ecosystem | | | Method | Reference | | |
|---|---|---|---|---|---|---|---|
| **Priest River Experimental Forestry, Idaho, US** | Temperate Forest | -116 | 48 | ARA | Harvey et al (1989) | 0.1 | 1.7 |
| **Arapaho Prarie, Nebraska, US** | Grassland | -100 | 42 | unspecified | Kaputsa and DuBois (1987) | 0.2 | 0.7 |
| **Lynx Prairie Preserve, Ohio, US** | Grassland | -83.5 | 39 | ARA | DuBois and Kaputsa(1983) | 8.2 | 1.9 |
| **Konza Prarie Research Natural Area, Kansas, US** | Grassland | -96 | 39.5 | nitrogenase activity | Eisele et al (1989) | 21 | 3.3 |
| **Buso, Papua New Guinea** | Tropical Forest | 147 | -7.5 | ARA | Goosem and Lamb (1986) | 0.5 | 5.2 |
| **Reserve Ducke, Manaus, Brazil** | Tropical Forest | -59 | -3 | ARA | Sylvester-Bradley et al. (1980) | 2.45 | 3.5 |
| **Sinharaja Man and Biosphere reserve** | Tropical Forest | 80.5 | 6.5 | ARA | Maheswaran and Gunatilleke (1990) | 8 | 8.5 |
| **Amazon Territory of Venezuela** | Tropical Forest | -67 | 2 | ARA | Jordan et al (1983) | 32 | 20.3 |
| **Kilauea, Hawaii, US** | Tropical Forest | -155 | 19 | ARA | Vitousek (1994) | 2.8 | 18.5 |
| **Volcano La Soufriere, Guadeloupe** | Tropical Forest | -61.5 | 16 | ARA | Sheridan (1991) | 4.02 | 7.2 |
| **Hawaii Volcanoes National Park, US** | Tropical Forest | -155 | 19.5 | ARA | Ley and D'Antonio (1998) | 4.9 | 9.3 |
| **Santa Ynez Mountain, California, US** | Mediterranean Shrubland | -120 | 34.5 | ARA | Schlesinger et al.(1982) | 1 | 2.4 |
| **San Bernardino Mountains, California, US** | Mediterranean Shrubland | -116.5 | 34 | ARA | Lepper and Fleschner (1977) | 6.9 | 3.7 |
| **Harpers Well, California, US** | Xeric Shrubland | -116 | 33.5 | N accumulation | Rundel et al.(1982) | 30 | 18.5 |
| **Sonoran Desert, Arizona, US** | Xeric Shrubland | -112.5 | 33 | cation accumulation | Jarrell and Virginia (1990) | 40 | 23.5 |

*ARA denotes the acetylene reduction assay method in determining biological $N_2$ fixation rates.

**Table 3.** Model estimated biological nitrogen fixation in global natural terrestrial ecosystems

| Ecosystem | Average coverage of $N_2$ fixing plants | Coverage range | Reference | $N_2$ Fixation Rate (kg N ha$^{-1}$ yr$^{-1}$) | Total_Min (Tg N yr$^{-1}$) | Total_Max (Tg N yr$^{-1}$) | Total_Avg (Tg N yr$^{-1}$) | Area ($10^8$ ha) |
|---|---|---|---|---|---|---|---|---|
| wet tundra | 9% | 3%~15% | May and Webber (1982) | 3.2 | 0.51 | 2.55 | 1.54 | 5.37 |
| alpine tundra & wet tundra | 9% | 3%~15% | May and Webber (1982) | 3.2 | 0.51 | 2.55 | 1.54 | 5.36 |
| boreal forest | 9% | 4%~18% | Alexander and Billington (1986); weber and Van Cleve (1981) | 2.1 | 2.01 | 9.06 | 4.53 | 19.3 |
| temperate coniferous forest | 5% | 1%~10% | Cleveland et al (1999) | 12.7 | 0.71 | 7.15 | 3.5 | 5.51 |
| temperate deciduous forest | 5% | 1%~10% | Cleveland et al (1999) | 12.7 | 0.76 | 7.65 | 3.75 | 5.89 |
| temperate evergreen forest | 5% | 1%~10% | Cleveland et al (1999) | 12.7 | 0.43 | 4.34 | 2.13 | 3.35 |
| grassland | 15% | 5%~25% | Woodmansee et al (1981); Robertson and Rosswall (1986) | 1.9 | 0.61 | 3.1 | 1.86 | 8.4 |
| tropical forest | 15% | 5%~25% | Cleveland et al (2001) | 18.2 | 10.8 | 54 | 32.6 | 17.8 |
| xeric shrubland | 15% | 10%~20% | Johnson and Mayeux (1990) | 5.7 | 2.92 | 14.6 | 8.35 | 14.8 |
| Mediterranean shrubland | 15% | 10%~20% | Johnson and Mayeux (1990) | 2.7 | 0.13 | 0.66 | 0.4 | 1.47 |
| savanna | 15% | 5%~25% | Stewart et al (1978); Bate and Gunton (1982) | 1.9 | 0.45 | 2.23 | 1.34 | 7.05 |
| **Total** | | | | | **19.84** | **107.89** | **61.54** | **94.3** |

**Table 4.** Model parameters for various natural terrestrial ecosystems

| | | N _pot (g N$_2$ fixed day $^{-1}$) | t_optL (°C) | t_optH (°C) | W_upH (J kg $^{-1}$) | fNup | Kc (g C m$^{-2}$) |
|---|---|---|---|---|---|---|---|
| 1 | wet tundra | 0.028 | 10 | 25 | 0.8 | 65 | 0.002 |
| 2 | alpine tundra & wet tundra | 0.028 | 10 | 25 | 0.8 | 65 | 0.002 |
| 3 | boreal forest | 0.032 | 12 | 25 | 0.8 | 70 | 0.008 |
| 4 | temperate coniferous forest | 0.55 | 16 | 35 | 0.6 | 80 | 0.01 |
| 5 | temperate deciduous forest | 0.55 | 18 | 35 | 0.6 | 80 | 0.01 |
| 6 | temperate evergreen forest | 0.55 | 18 | 35 | 0.6 | 80 | 0.01 |
| 7 | grassland | 0.05 | 18 | 35 | 0.5 | 60 | 0.012 |
| 8 | tropical forest | 0.8 | 20 | 35 | 0.8 | 100 | 0.005 |
| 9 | xeric shrubland | 0.7 | 15 | 35 | 0.4 | 65 | 0.016 |
| 10 | Mediterranean shrubland | 0.08 | 19 | 35 | 0.5 | 65 | 0.016 |
| 11 | savanna | 0.05 | 20 | 35 | 0.5 | 60 | 0.012 |

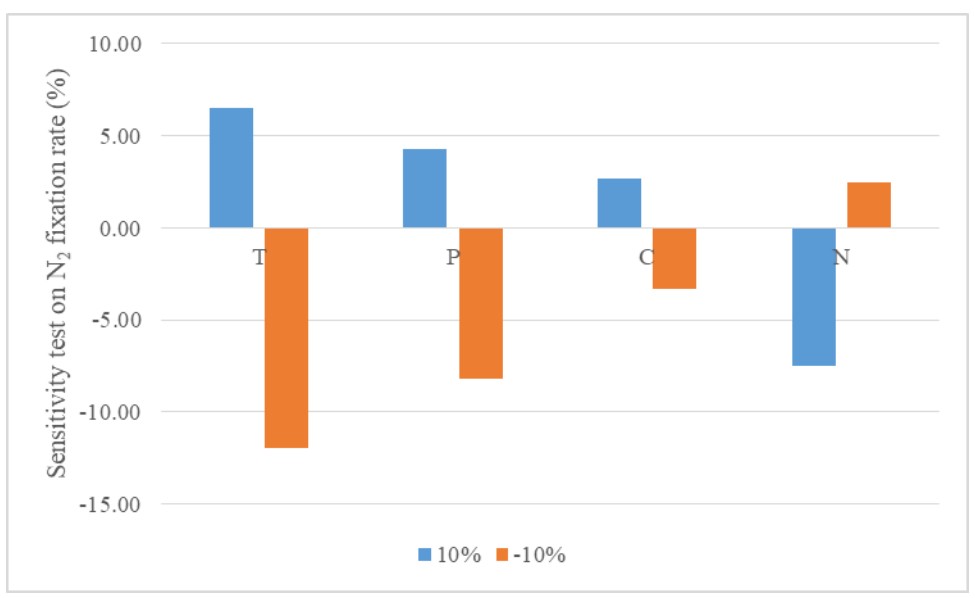

**Figure 1.** Model sensitivity of $N_2$ fixation in natural terrestrial ecosystems to changing model input data: Increasing or decreasing each variable by 10% for air temperature (T), precipitation (P), soil carbon content (C), soil nitrogen content (N) for $N_2$ fixation rate.

(a)

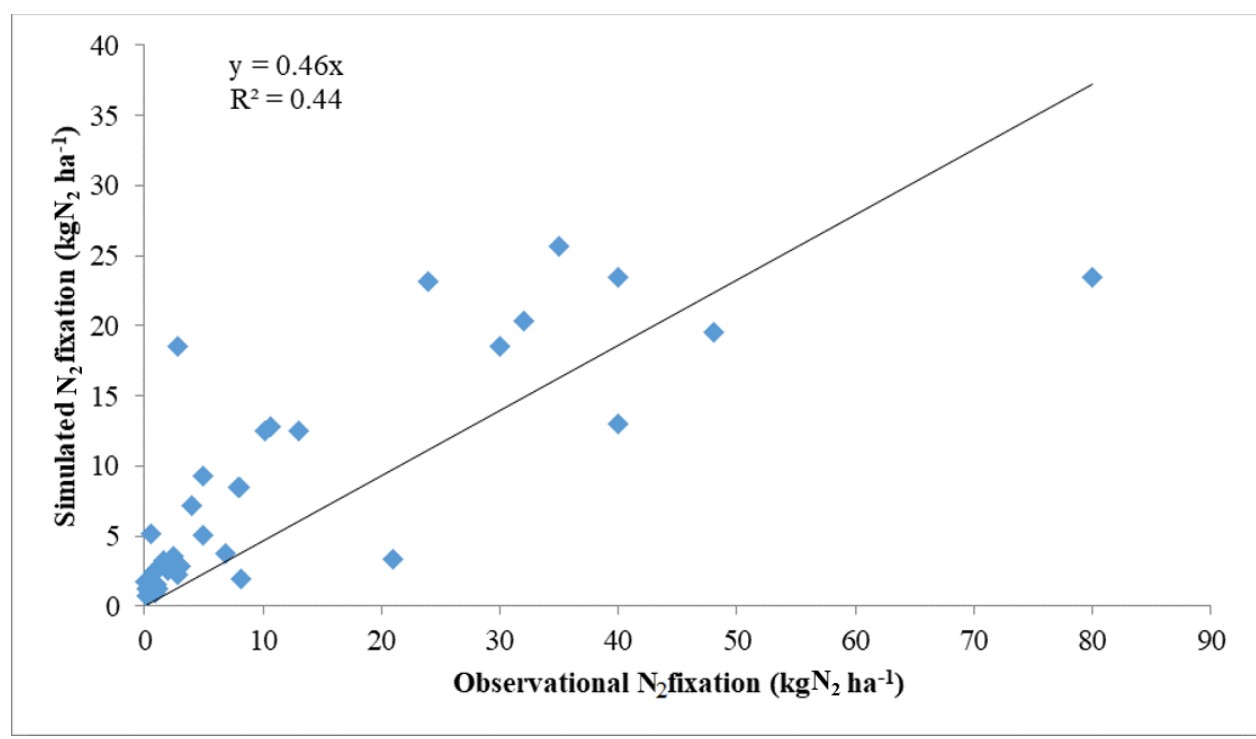

(b)

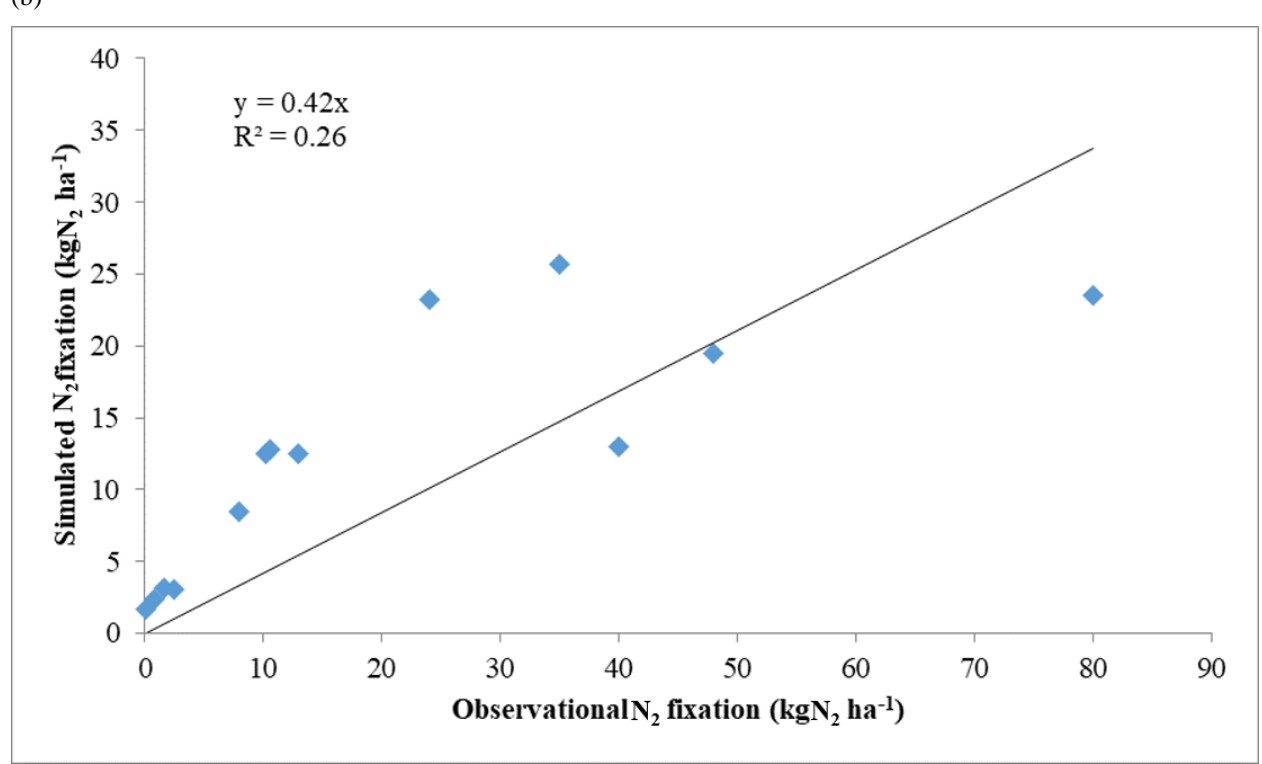

(c)

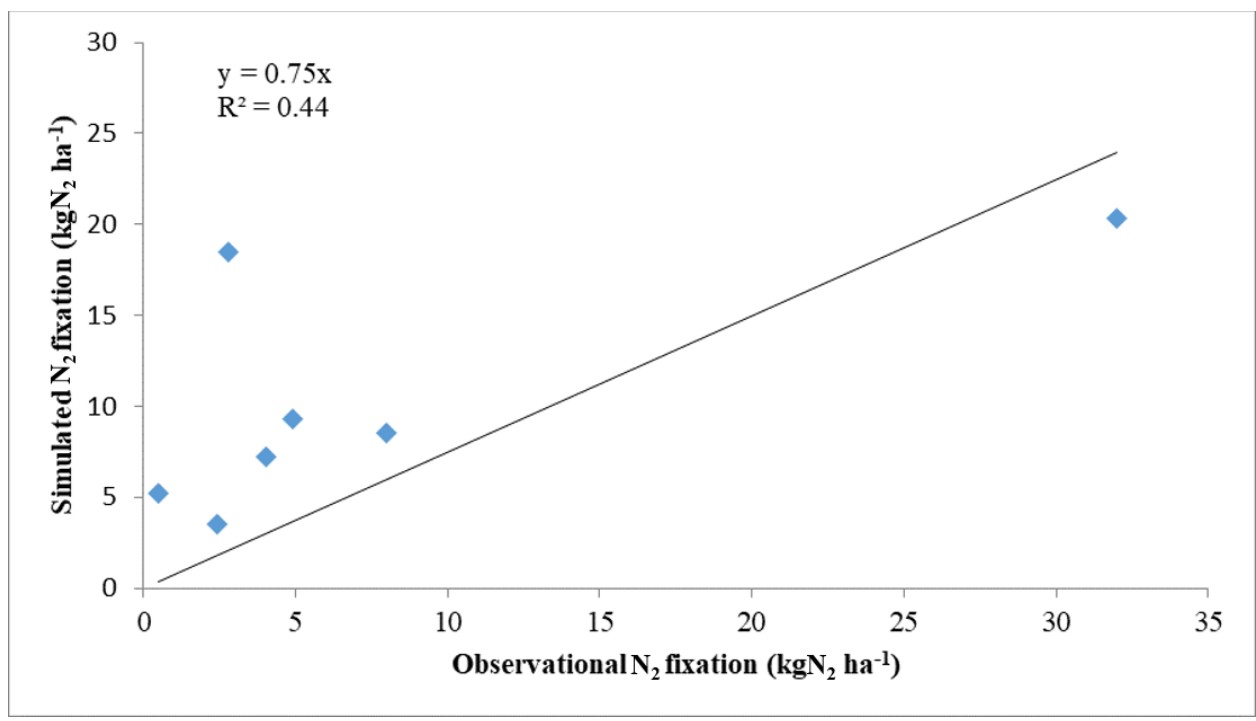

**Figure 2.** Comparison between Modeled and Observed Nitrogen Fixation Rate at site level: (a) All sites, (b) Temperate Forest, (c) Tropical Forest (data listed in Table 2). Y is simulated $N_2$ fixation while X represents the observational $N_2$ fixation.

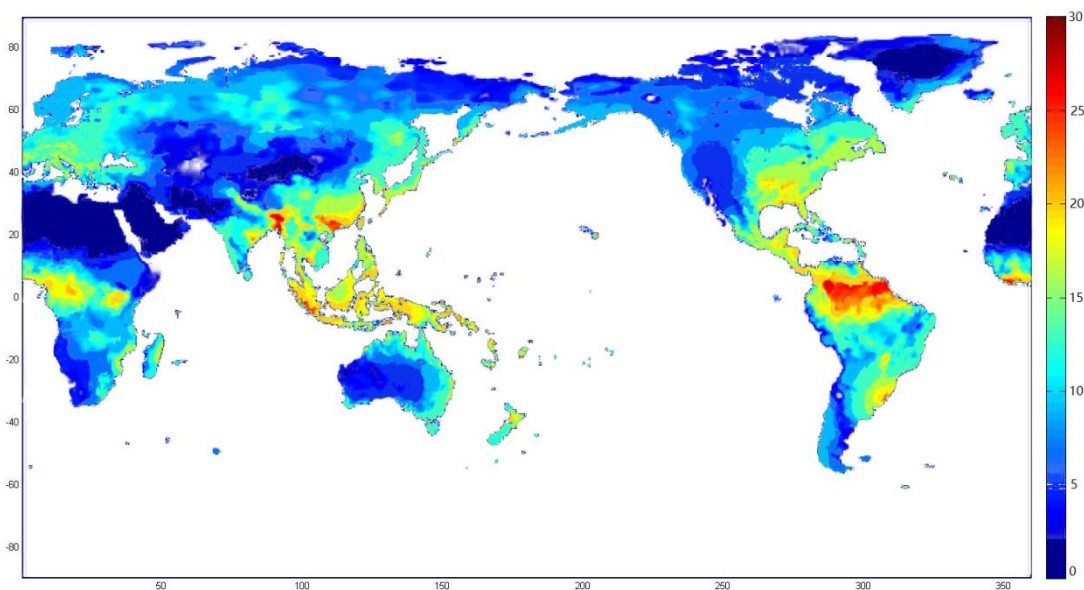

**Figure 3.** Simulated spatial distribution of BNF rates (kg $N_2$ ha$^{-1}$ yr$^{-1}$) in natural terrestrial ecosystems during 1990-2000 by considering the BNF effects.