# Peer review of "Modelling biological nitrogen fixation in global natural terrestrial ecosystems"

_Biogeosciences, 2019_

## Referee Comment (RC1) · Anonymous Referee #1 · 22 Aug 2019

General comments

Yu and Zhang present an interesting approach to address an ongoing gap in our collective knowledge: we simply do not know how much nitrogen is being fixed on earth. The range of estimates for this process remain very large, and approaches to improve the accuracy of these estimates are necessary.

As such, efforts such as this modeling approach, are welcomed additions to the debate. This manuscript lays out a modeling approach focused on the contribution of legumes

to global rates of nitrogen fixation. They estimate that globally nitrogen fixation introduces $\sim$ 61 Tg N yr-1 into terrestrial ecosystems. This number is on the lower order of experimental scaling compendiums, however, well within the bounds of estimations made on the basis of isotope box models. However, as this manuscript points out, there are significant pathways of N-fixation not represented by this model (free-living fixation via heterotrophic bacteria or cryptogamic crusts) that could contribute significantly to terrestrial fixation, which would presumably increase their mean estimate towards that calculated via the relationship between N-fixation and evapotranspiration.

Overall, I think this is a nice manuscript, and a good contribution to this debate. However, I had a hard time following some modeled controls on N-fixation (see my comments), and was also confused by the purported relationship between N-fixation and nitrous oxide production. I could be wrong, but there is not, to my knowledge, a linear relationship between natural fixation and nitrous oxide production. It's clear that areas with high fixation are also sometimes areas of high N2O production, however, this is more to do with conditions at the site than a direct link? Furthermore, N2O production is a far more episodic process than N-fixation. Therefore point measurements of N2O can represent significant underestimates, or overestimates, of annual N2O production depending on the conditions that measurements took place under. Add to this the role of nutrient use efficiency in retaining or releasing nutrients within or from plant biomass, which varies depending on species and stand age. For these reasons, it seems problematic to compare N-fixation with N2O production, and I would ask for a expanded justification for this aspect of the study.

I also think the title is somewhat misleading. As I see it, the relationship between fixation and N2O production is not the central feature of this manuscript. This manuscript describes a new module within an existing model, and the factors controlling the rate of fixation. The majority of the discussion is concerned with comparing the estimates generated here with estimates from previous studies.

Another issue I found was that estimates for fixation span from the tropics to arctic

regions, yet the model itself does not represent the dominant fixers within arctic regions. While I don't doubt the generalized findings that the arctic regions have lower fixation than tropical regions, it seems that the model cannot accurately estimate fixation rates for Arctic regions with it's current structure leaving me to wonder how to interpret the values reported for Arctic regions?

Finally, while the quantitative nature of the manuscript is appreciated, the quality of the writing is quite poor. A few examples are given below, however, I believe prior to resubmission the authors should correct the grammatical mistakes throughout. In general, the writing needs to be tightened up throughout too.

Specific comments

The above mentioned grammatical issues include, but are not limited to,

Ln 30: This sentence needs to be tightened up, but the second half could be changed from, 'and decrease from the equator', to 'which decreases from. . .'.

Ln 34: Remove 'the' prior to 'fixation'.

Ln 36: Change 'types', to 'type'.

Ln 36: Remove 'the' before 'biological nitrogen fixation'.

Other examples are apparent throughout the manuscript (e.g., Ln. 97; Ln 100, Ln 109, Ln 256, etc.).

Ln 51: Gruber and Galloway, 2008, Nature, 451(17), 293-296, would help constrains the quantitative aspect of this sentence.

Ln 80: I think it would help to have a model schematic here - I understand a schematic of the broader model has been published previously, but repeating that schematic and focusing on the newly integrated processes would help the reader.

Ln 112: What data is required to derive these estimates?

Equation 4: What is the origin of the soil nitrogen inhibition values? Please reference the manuscript these values were taken from.

Ln 174: I think the authors mean 'a priori values', no?

Ln 226; Should this value be -5 % rather than 5 %? I think the value of -5 % is reported elsewhere in the manuscript.

Ln 229: What controls N2O fluxes within the model?

Ln 297: What does 'affected' mean here? It would help to be more specific about direction, for example, does it mean enhanced or reduced?

Ln 368: Change subheading 'Major controls to. . .', to, 'Major controls on. . .'.

Table 5 needs references to explain the origin of and biome specific variability in these values.

Seems like Fig. 1 is a repetition of data in the tables? Is Fig. 1 needed? Particularly because there is far more information in the tables.

Please improve the quality of the figure 2, which is quite poor. Why are the x-axis values just floating in the middle of the figure? And why abbreviate them?

Technical comments.

Ln 84: It's not clear to me if the published ARA studies used to calibrate the model have all themselves been calibrated with 15N measurements? The ARA approach is notoriously difficult to interpret without reliable calibration, and the conversion of acetylene reduced to nitrogen fixation ranges significantly depending on various factors include the specific nitrogenase enzyme.

Ln 124: I'm confused by this temperature relationship, is this a Gaussian distribution similar to that laid out by Houlton et al., (Nature, 2008, 454, 327-330)? It doesn't appear to be - this relationship sounds like there is a very broad plateau whereby temperature

does not limit fixation across a wide range (12 - 35 C). Is this correct?

---

## Referee Comment (RC2) · Anonymous Referee #2 · 20 Dec 2019

This paper describes the development of a process-based model to simulate global nitrogen fixation in terrestrial soils and then couples that model with previously published model(s) predicting N dynamics (Terrestrial Ecosystem Model, TEM) and N2O emissions from soils. The goal was to quantify the contribution of biological nitrogen fixation (BNF) to global N2O emissions and to better match the simulated emissions with observed data. The BNF calculation exercise is interesting and probably a worthwhile contribution to the N general model. However the major focus on tying that to N2O emissions is more problematic and the paper could benefit from some redirection.

Some issues that need to be addressed concerning assumptions made, methodology, conclusions, and clarity of presentation.

1) The term 'natural' is frequently used throughout the text without attempting to define the context in which it is intended. 'Natural' has many implications, connotations, and hidden assumptions. An explicit definition of the term, as it is used on the paper, is needed. How well do the field sites selected relate to that definition? Indeed, since estimates are that humans have doubled the amount of fixed N applied to the terrestrial landscape of the planet (see the Galloway et al., 2004 reference), how does that relate to the 'natural' sites? More importantly, how can global extrapolation of BNF relate to global N2O emissions given the substantial contribution that the anthropogenic fixed N (ANF) must have made? While the paper does not completely ignore the importance of ANF, it does not make clear distinctions between the two relative to N2O emissions.

2) The paper only mentions the TEM and the N2O emissions model in passing. Little information is provided as to how the BNF model is integrated into those pre-existing models to derive N2O emissions. Where does the newly fixed N enter into those models? Was it considered to increase the soil organic-N pool size? Or was the assumption made that newly fixed N was all immediately taken up by plants, or both? Were other parameters, such as soil inorganic-N supply, in the previous TEM model modified when the BNF model was included?

3) The general conclusion is that including BNF resulted in additional N that led to -5% to +20% changes in seasonal soil N2O emissions. The main differences occurred in the winter months. That was the range, but what was the central tendency of the effect? Figure 5 suggests that there was generally little change in emissions overall. Indeed, one could easily conclude from Figure 5 that including BNF in the larger model did not have a substantial impact on N2O emissions. Perhaps that is not too surprising when one considers that the total fixed N pool size (plant biomass + soil fixed N + atmospheric input) must be substantially larger than the annual amount of newly fixed N from BNF. The abstract states that: "This study highlights that there are relatively

large effects of the biological nitrogen fixation on ecosystem nitrogen cycling and soil N2O emissions." The results shown in the paper and the discussion do not at all agree with that conclusion relative to N2O emissions.

4) Given the large overlap in tables and figures between this paper and Yu and Zhuang 2019, one wonders whether the incremental contribution of this paper relative to N2O emissions represents a publishable, stand-alone contribution over and above Yu and Zhuang 2019. How does this differ from a laboratory experiment that adds little to no new insight into what is already known? In my opinion, the paper can be, and should be, strengthened, by including additional considerations, such as ANF or the differential effect(s) of N speciation on BNF. Or perhaps more radically by reducing or eliminating the focus on N2O emissions all-together and refocusing on how the BNF inclusion changes the N cycle fluxes in the TEM model. In short, the paper has too much emphasis on N2O emission given the Yu and Zhuang 2019, paper while more could be done overall concerning the BNF contribution to the model.

5) Can N2O emission data from 8 (line 170), or 6 (line 197), or 5 (Table 3) sites (which is it?) be reasonably extrapolated globally? Those sites were chosen because they were "affected by legumes." What is the implication of that to the extrapolation? Yes, it was subsequently tested on 35 other (?) sites. But there was little N fixation measured for almost half of those sites.

6) The paper would benefit greatly from careful editing. There are numerous errors and discrepancies, particularly between the text and the figures and tables. Many are listed below.

Other, more minor comments:

Line 63. The EPA reference is missing.

Line 104: "and for spatial limitation". How does that relate to C demand?

Eqn 1: Nfix is not defined in the text.

Line 143: use the same terminology throughout the paper. Upper threshold is given here. In the table it listed as 'upper bond' (sic).

Line 155: change to read "every unit of nitrogen fixed..."

Line 161: the units for Cr in the text do not match the units given in Table 1.

Line 170: 8 sites are indicated, but only 5 are listed in table 3.

Line 186 and throughout: use past tense.

Line 186: Table 2 lists 7 ecosystem types, not 11.

Line 197: Table 3 lists 5 field sites, not 6.

Line 201: should be "sensitivity testing."

Line 205: Figure 2 is a result, not a method.

Line 214: no standard deviation is given.

Line 214: What is the rationale for "removing these data?"

Lines 216-217: "simulations are closer . . .in temperate forests. . ." Close to what?

Line 228: There is no N2O data in Figure 3.

Ines 244-247: Sentence starting with "Here" is unclear. Is that referring to the previously cited study or this study? Usually that term refers to the current study, but it appears to be referring to Bruijnzeel et al.

Lines 229-231: "The comparison between measured and simulated data further shows the influence of BNF for different ecosystem types. . ." This is unclear and needs further explanation.

Line 248: 32.5 Tg N does not match the number in Table 4.

Line 248-249: give the numbers in text and refer to Table 4.

[Figure]

Line 279-281: This sentence should include some mention of the high nitrate concentrations typically found in desert soils.

Lines 297-298: What was the overall mean and standard deviation of the model results when BNF was and was not included in the N2O emission simulations?

Line 374: change slightly to slight.

Lines 381-384: Not all agree with this statement. See Heden et al., 2009, Ann. Rev. Ecol. Evol. Syst. 40:613. Alternate explanations should be included here.

Table 1: Change bond to bound throughout. Provide units for all parameters and ranges for coefficients. The description for the Michaelis-Menton constant is incomplete. What process is that a constant for?

Table 2: Column headers should be "measured N fixation rate" and "simulated N fixation rate."

Table 3: Wagga Wagga is in Australia.

Table 5: N_pot parameter format and units do not match Table 1. What are units for fNup? Units for Kc do not match Table 1.

Figure 1: The grey to blue colors are hard to distinguish against the green and blue background shading. Use the same units for N2O emissions and N fixation rate here and throughout the paper.

Figure 3: What do the lines represent? Regression lines forced through zero? If so, what is the rationale for doing that?

Figure 5: Point out the y-axis scale differences. The scales chosen for the two tropical forests is rather misleading and are based on what appear to be outlier observations. Suggest using the same scale with a y axis break to include the outliers so that these two panels can be more easily compared. What ecosystem is panel e?

[Figure]

---

## Author Response (AR1)

Response letter-1

General comments:

1. …For these reasons, it seems problematic to compare N-fixation with N2O production, and I would ask for a expanded justification for this aspect of the study.
   *Response: After a careful consideration on the problem, we think it need further work to build a solid convincing relation between natural BNF and N2O emissions. Thus, in the revision, we will focus on modeling BNF only.*

2. I also think the title is somewhat misleading…..
   *Response: Because of the adjustment in content, we will give a new title. An tentative title is "Modeling biological nitrogen fixation in global natural terrestrial ecosystems"*

3. …it seems that the model cannot accurately estimate fixation rates for Arctic regions with it's current structure leaving me to wonder how to interpret the values reported for Arctic regions?
   *Response: More observational data in Arctic area will be helpful for a more reliable simulation. Currently, it is not so convincing as that from other ecosystem types. We will try our best to search for more on-site data and related works, for further interpretation of the modeled fixation in arctic regions.*

4. Grammatical mistakes
   *Response: Thank you for your suggestion. We will correct these.*

Minor revisions:

- Ln30: This sentence needs to be tightened up, but the second half could be changed from, 'and decrease from the equator', to 'which decreases from: : :'.
  *Corrected.*

- Ln 34: Remove 'the' prior to 'fixation'.
  *Corrected.*

- Ln 36: Change 'types', to 'type'.
  *Changed*

- *Ln 36: Remove 'the' before 'biological nitrogen fixation'.*
  Changed.

- Ln 51: Gruber and Galloway, 2008, Nature, 451(17), 293-296, would help constrains the quantitative aspect of this sentence.
  *Thank you for your suggestion. we have added information related to anthropogenic nitrogen into this sentence.*

- Ln 80: I think it would help to have a model schematic here - I understand a schematic of the broader model has been published previously, but repeating that schematic and focusing on the newly

integrated processes would help the reader.

*Please refer to Figure 1 from Yu and Zhuang (2019). The model schematic is very similar to this one, if not the same.*

- Ln 112: What data is required to derive these estimates?
*Please refer section 2.3 to the information of data.*

- Equation 4: What is the origin of the soil nitrogen inhibition values? Please reference the manuscript these values were taken from.

*$N_s$ is soil mineral nitrogen, which is a variable calculated in our model. $f_{Nup}$ is a parameter related to legume nitrogen fixation. Its value and reference manuscript can be found in Table 1.*

- Ln 174: I think the authors mean 'a priori values', no?
*Yes, it is.*

- Ln 226; Should this value be -5 % rather than 5 %? I think the value of -5 % is reported elsewhere in the manuscript.
*Thank you for your careful reading. It should be -5% here. we have corrected this part.*

- Ln 229: What controls N2O fluxes within the model?
*You can find every detail about the controls of N2O fluxes from Yu and Zhuang (2019). But actually, we will remove the discussion on N2O emissions in revision.*

- Ln 297: What does 'affected' mean here? It would help to be more specific about direction, for example, does it mean enhanced or reduced?
*We have clarified this sentence. It is also explained in the rest of this paragraph.*

- Ln 368: Change subheading 'Major controls to: : :', to, 'Major controls on: : :'.
*Changed*

- Seems like Fig. 1 is a repetition of data in the tables? Is Fig. 1 needed? Particularly because there is far more information in the tables.
*It is a straightforward visualization with the location information of Table 1. We will consider whether to keep it after the adjustment of text content.*

- Please improve the quality of the figure 2, which is quite poor. Why are the x-axis values just floating in the middle of the figure? And why abbreviate them?
*We will replot the table. Abbreviations are used here because the original ones are too long.*

Technical comments:
- Ln 84: It's not clear to me if the published ARA studies used to calibrate the model have all themselves been calibrated with 15N measurements? The ARA approach is notoriously difficult to

interpret without reliable calibration, and the conversion of acetylene reduced to nitrogen fixation ranges significantly depending on various factors include the specific nitrogenase enzyme.

*We will add a brief introduction about the published methods.*

- Ln 124: I'm confused by this temperature relationship, is this a Gaussian distribution similar to that laid out by Houlton et al., (Nature, 2008, 454, 327-330)? It doesn't appear to be - this relationship sounds like there is a very broad plateau whereby temperature does not limit fixation across a wide range (12 - 35 C). Is this correct?

*This paragraph gives some examples on the temperature influence on different vegetation types. It is not a Gaussian distribution. We will add some explanation in this part to avoid confusions.*

Response letter-2

*Response: Thank you for the overall positive feedback. We have thoroughly revised the paper following your comments and suggestions. We find that most problems focused on the relationship between natural N fixation and N2O emissions. Because it is difficult to avoid the influence of anthropogenic fixed nitrogen on natural ecosystem N2O emissions at this time. We decide, in the revision, to focus on the simulation of N fixation and eliminate the evaluation of fixation impacts on N2O emissions.*

1) The term 'natural' is frequently used throughout the text without attempting to define the context in which it is intended. 'Natural' has many implications, connotations, and hidden assumptions. An explicit definition of the term, as it is used on the paper, is needed. How well do the field sites selected relate to that definition? Indeed, since estimates are that humans have doubled the amount of fixed N applied to the terrestrial landscape of the planet (see the Galloway et al., 2004 reference), how does that relate to the 'natural' sites? More importantly, how can global extrapolation of BNF relate to global N2O emissions given the substantial contribution that the anthropogenic fixed N (ANF) must have made? While the paper does not completely ignore the importance of ANF, it does not make clear distinctions between the two relative to N2O emissions.

   *Response:*
   - *In this study, we only considered natural ecosystem emissions. Croplands emissions were not modeled. We will add a clear definition of natural in the introduction part.*
   - *We cannot guarantee there is absolutely no human effects on selected sites. But according to the cited paper, they are treated as natural environment.*
   - *Thank you for your suggestion on ANF's influence on N2O emissions. We have eliminated N2O results, and only present the N fixation results.*

2) The paper only mentions the TEM and the N2O emissions model in passing. Little information is provided as to how the BNF model is integrated into those pre-existing models to derive N2O emissions. Where does the newly fixed N enter into those models? Was it considered to increase the soil organic-N pool size? Or was the assumption made that newly fixed N was all immediately taken up by plants, or both? Were other parameters, such as soil inorganic-N supply, in the previous TEM model modified when the BNF model was included?

   *Response:*
   - *We will only simulate the N fixation part with the assistant of TEM.*

3) The general conclusion is that including BNF resulted in additional N that led to -5% to +20% changes in seasonal soil N2O emissions. The main differences occurred in the winter months. That was the range, but what was the central tendency of the effect? Figure 5 suggests that there was generally little change in emissions overall. Indeed, one could easily conclude from Figure 5 that including BNF in the larger model did not have a substantial impact on N2O emissions. Perhaps that is not too surprising when one

considers that the total fixed N pool size (plant biomass + soil fixed N + atmospheric input) must be substantially larger than the annual amount of newly fixed N from BNF. The abstract states that: "This study highlights that there are relatively large effects of the biological nitrogen fixation on ecosystem nitrogen cycling and soil N2O emissions." The results shown in the paper and the discussion do not at all agree with that conclusion relative to N2O emissions.

*Response:*
- *We will modify the content of Figure 5.*
- *This study will not highlight the relation between N fixation and N2O emissions.*

4) Given the large overlap in tables and figures between this paper and Yu and Zhuang 2019, one wonders whether the incremental contribution of this paper relative to N2O emissions represents a publishable, stand-alone contribution over and above Yu and Zhuang 2019. How does this differ from a laboratory experiment that adds little to no new insight into what is already known? In my opinion, the paper can be, and should be, strengthened, by including additional considerations, such as ANF or the differential effect(s) of N speciation on BNF. Or perhaps more radically by reducing or eliminating the focus on N2O emissions all-together and refocusing on how the BNF inclusion changes the N cycle fluxes in the TEM model. In short, the paper has too much emphasis on N2O emission given the Yu and Zhuang 2019, paper while more could be done overall concerning the BNF contribution to the model.

*Response:*
*Thank you for your suggestion. We decide to focus our discussion on BNF in the revised version.*

5) Can N2O emission data from 8 (line 170), or 6 (line 197), or 5 (Table 3) sites (whichis it?) be reasonably extrapolated globally? Those sites were chosen because they were "affected by legumes." What is the implication of that to the extrapolation? Yes, it was subsequently tested on 35 other (?) sites. But there was little N fixation measured for almost half of those sites

*Response:*
*We will carefully deal with the observational data for N fixation in revision. The N2O emission data will be removed from the paper.*

6) Minor revision
- Line 63. The EPA reference is missing.
  *Added*

- Line 104: "and for spatial limitation". How does that relate to C demand?
  *Deleted "and for spatial limitation"*

- Eqn 1: Nfix is not defined in the text.
  *Added the definition in the text above Eqn 1.*

- Line 143: use the same terminology throughout the paper. Upper threshold is given
  here. In the table it listed as 'upper bond' (sic).
  *We have changed the "upper bond" to "upper threshold" in the table.*

- Line 155: change to read "every unit of nitrogen fixed..."
  *Added "fixed".*

- Line 161: the units for Cr in the text do not match the units given in Table 1.
  *Changed the unit in Table 1.*

- Line 170: 8 sites are indicated, but only 5 are listed in table 3.
  *Table 3 will be deleted.*

- Line 186 and throughout: use past tense.
  *Done.*

- Line 186: Table 2 lists 7 ecosystem types, not 11.
  *Changed to "7 ecosystem types among 11".*

- Line 197: Table 3 lists 5 field sites, not 6.
  *Table 3 will be deleted.*

- Line 201: should be "sensitivity testing."
  *Done.*

- Line 205: Figure 2 is a result, not a method.
  *Deleted the citation of Figure 2.*

- Line 214: no standard deviation is given.
  *Added.*

- Line 214: What is the rationale for "removing these data?"
  *Because they can be viewed as outliers of observation data.*

- Lines 216-217: "simulations are closer : : :in temperate forests: : :" Close to what?
  *Closer to observation.*

- Line 228: There is no N2O data in Figure 3.
  *We will delete the N2O part.*

- Line 244-247: Sentence starting with "Here" is unclear. Is that referring to the

previously cited study or this study? Usually that term refers to the current study, but it appears to be referring to Bruijnzeel et al.
*Changed "Here" to "In this study".*

- Lines 229-231: "The comparison between measured and simulated data further shows the influence of BNF for different ecosystem types: : :" This is unclear and need further explanation.
  *This sentence will be deleted.*

- Line 248: 32.5 Tg N does not match the number in Table 4
  *Changed the value in the text.*
  .
- Line 248-249: give the numbers in text and refer to Table 4.
  *Done.*

- Line 279-281: This sentence should include some mention of the high nitrate concentrations typically found in desert soils.
  *We will add a sentence discussing the nitrate concentration in desert.*

- Lines 297-298: What was the overall mean and standard deviation of the model results when BNF was and was not included in the N2O emission simulations?
  *We will delete the discussion on N2O emissions.*

- Line 374: change slightly to slight.
  *Changed.*

- Lines 381-384: Not all agree with this statement. See Heden et al., 2009, Ann. Rev. Ecol. Evol. Syst. 40:613. Alternate explanations should be included here.
  *Thank you for providing reference. We will refer to the reference to include other statements.*

- Table 1: Change bond to bound throughout. Provide units for all parameters and ranges for coefficients. The description for the Michaelis-Menton constant is incomplete. What process is that a constant for?
  *We have changed "bond" to "bound" in the description.*
  *Some of coefficients have no unit, but we have added units for those who have.*
  *The Michaelis-Menton constant is for Eqn.5, the consideration of soil carbon.*

- Table 2: Column headers should be "measured N fixation rate" and "simulated N fixation rate."
  *Changed.*

- Table 3: Wagga Wagga is in Australia.
  *Table 3 will be deleted.*

- Table 5: N_pot parameter format and units do not match Table 1. What are units for fNup? Units for Kc do not match Table 1.

  *We have unified the format and units between Table 1, 5 and in the text.*

- Figure 1: The grey to blue colors are hard to distinguish against the green and blue background shading. Use the same units for N2O emissions and N fixation rate here and throughout the paper.

  *We have unified the units between Figure 1 and text.*

- Figure 3: What do the lines represent? Regression lines forced through zero? If so, what is the rationale for doing that?

  *Yes, the lines are regression lines forced through zero. The inner rationale is explained in the text.*

- Figure 5: Point out the y-axis scale differences. The scales chosen for the two tropical forests is rather misleading and are based on what appear to be outlier observations. Suggest using the same scale with a y axis break to include the outliers so that these two panels can be more easily compared. What ecosystem is panel e?

  *Figure 5 will be removed in revision.*

[revised manuscript text omitted]

<s>This study first models the BNF from the symbiotic relationship between legume plants and bacteria. The model is then coupled with an existing N₂O biogeochemistry to quantify the BNF impact on N₂O emissions from global terrestrial ecosystem soils.</s>

**2. Methods**

**2.1 Overview**

We first develop a BNF model and then couple the model with an earlier version of <s>N₂O</s> biogeochemistry model quantifying soil carbon and nitrogen dynamics (Yu and Zhuang, 2019). The revised model is then used to quantify the BNF at regional and<s>the</s> global scale <s>and its impacts on soil N₂O emissions from</s> in natural terrestrial ecosystems. The BNF rate estimates consider the effects of environmental conditions including temperature, soil moisture, soil mineral nitrogen content and soil carbon content. The <s>new</s> modified model is calibrated and evaluated with observed N fixation rate data from published studies for various natural terrestrial ecosystems from the Arctic to tropical ecosystems. The model sensitivity to model input is analyzed. The model is then extrapolated to the global terrestrial ecosystems at a monthly step and a spatial resolution of 0.5° by 0.5° for the final decade of the 20$^{th}$ century. The effects of physical conditions on BNF are then analyzed.<s>The effects of BNF and atmospheric nitrogen depositions on N₂O emissions from natural terrestrial ecosystem soils are analyzed.</s>

[revised manuscript text omitted]
 evaluate the model's performance on capturing seasonal trends, the simulated N$_2$O emissions were plotted along with observational data for the observational period. A linear regression is conducted to show the similarity and discrepancy between simulations and observations, R², along with the slope and intercept of the linear regression line are calculated. Six field observational sites having legume were organized for the model validation (Table 3). Monthly N₂O emission data were used for model comparison.

210    **2.5 Model Sensitivity and Uncertainty Analysis**

The response of N fixation and N₂O emissions of different biomes to input data, and variation of parameters was analyzed using sensitivity test. Four major input variables were selected, including air temperature, precipitation, soil nitrogen content and soil organic carbon content. The monthly average input variables were changed by $\pm10\%$ of the original level for each site and each grid. The variables were changed at 6 levels,
215    respectively, and the rest of input variables were kept at their original values. The sensitivity was calculated by comparing the simulated annual nitrogen fixation to the simulations with the original input values.
The sensitivity was calculated by comparing the simulated annual N₂O emissions and nitrogen fixation to the simulations with the original input values (Figure 2).

**3. Results**

220    **3.1 Model evaluation**

To evaluate the model, thirty-five observational sites were selected for 7 major ecosystem types across the globe, representing different climate and soil conditions. The experimental data of N fixation have a mean value of 12.9 kg N ha$^{-1}$ yr$^{-1}$, with a standard deviation of 17.7 kg N ha$^{-1}$ yr$^{-1}$. The maximum observed fixation occurred in

temperate forest in New Zealand, while the minimum rate was also for temperate forest in Idaho State of the US.

225 Our simulations are comparable with the observed data for all major ecosystem types with the coefficient of determination ($R^2$) of 0.44 and with a slope of 0.46 (Figure 2). The regression results are mainly influenced by some observed data greater than 30 kg N ha$^{-1}$ yr$^{-1}$ . By removing the outliers of observational data, the slope of regression increases to 0.72. Observational data for temperate forests show the greatest variation among all major ecosystem types, with a maximum value reaching 800 times of the minimum one.

230 Simulations are closer to the observations across sites in temperate forests with $R^2$ of 0.26 and slope of 0.42. Our model underestimated nitrogen fixation rate in temperate forests. The large variation in observations may be due to the distribution of legume plants, different sampling time periods (e.g., growing and non-growing seasons), and varying climate conditions. For tropical forests, our model estimates of N fixation are higher than observations with the slope of 0.75 and $R^2$ of 0.44.

235 ~~The model was also tested by comparing the simulated N$_2$O emissions with observed data for 5 sites and 3 ecosystem types including tropical forests, temperate forests, and grasslands. All observed data were converted into monthly average values. Our model reasonably reproduced the seasonal variations of N$_2$O emissions at observational sites. In our earlier version of the model, biological nitrogen input was assumed to be a constant throughout the year (Yu and Zhuang 2019). Compared to our earlier version, the current version contributes ~5% to~~

240 ~~20% of the total N$_2$O emission, but only leads to a minor difference to total seasonal trend throughout the year (Figure 3). There are still some discrepancies between simulated and observed N$_2$O emissions, which could be due to the uncertainty in measurement, sudden weather events and changes in soil characteristics. The comparison between measured and simulated data further shows the influence of BNF for different ecosystem types on N$_2$O emissions. The influence is larger in tropical areas, while by percentage, BNF is more important for grasslands in~~

245

**3.2 Model sensitivity analysis**

The model sensitivity analysis quantifies the impact of changes in forcing data on nitrogen fixation rate . Climate conditions including air temperature and precipitation, and soil characteristics of nitrogen content and carbon content varied at 3 levels to examine the sensitivity. The response of nitrogen fixation rate

[revised manuscript text omitted]

Spatially, the highest rate of N fixation occurred in the tropical and sub-tropical areas, as a result of proper climate and soil characteristics for fixers (Figure 3). N fixation from tropical forests and xeric shrubland contributes to nearly half of the global terrestrial amount (Table 3). A lower N fixation rate was in high latitudes of East China, North America and Europe, which were mainly covered with temperate forests. Compared to tropical areas, N fixation in temperate regions shows a larger variability depending on vegetation types. The spatial variation could be attributed to the distribution of legume plants, in addition to the difference of humidity and temperature conditions. N fixation in temperate regions accounts for 35% of the total fixed N.
* * *
~~The model was run at site level to estimate the role of BNF on N₂O emissions. Our analysis between current model and previous model without N fixation module indicated that N₂O emissions would be affected by -5% to 20% depending on biome types and seasons (Figure 5). In most cases, we got higher estimation of N2O emissions because BNF process enhances soil N content. The involvement of BNF strengthens the seasonal variation as BNF led to lower emissions in winter and higher emissions in summer.~~

[revised manuscript text omitted]

Mosier, A. R., Parton, W. J., Valentine, D. W., Ojima, D. S., Schimel, D. S., & Heinemeyer, O. CH4 and N2O fluxes in the Colorado shortgrass steppe: 2. Long-term impact of land use change. Global Biogeochem. Cy., 11(1), 29-42, 1997..

Mus, F., Crook, M. B., Garcia, K., Costas, A. G., Geddes, B. A., Kouri, E. D., ... & Udvardi, M. K. Symbiotic nitrogen fixation and the challenges to its extension to nonlegumes. Appl. Environ. Microbiol., 82(13), 3698-3710, 2016.

Müller, C., & Sherlock, R. R.. Nitrous oxide emissions from temperate grassland ecosystems in the Northern and Southern Hemispheres. Global Biogeochem. Cy., 18(1), 2004.

NOAA (National Oceanic and Atmospheric Administration). 2016. Monthly mean N2O concentrations for Barrow, Alaska; Mauna Loa, Hawaii; and the South Pole. Accessed June 8, 2016.

[revised manuscript text omitted]

(a)

[Figure]

(b)

[Figure]

**Figure 21.** Model sensitivity of N fixation and $N_2O$ emissions in natural terrestrial ecosystems to changing model input data:  Increasing or decreasing each variable by 10% for air temperature (T), precipitation (P), soil carbon content (C), soil nitrogen content (N) for (a) N fixation rate. and (b) N₂O emissions

(a)

[Figure]

(b)

[Figure]

(c)

[Figure]

**Figure 13.** Comparison between Modeled and Observed Nitrogen Fixation Rate at site level: (a) All sites, (b) Temperate Forest, (c) Tropical Forest (data listed in Table 2). Y is simulated N fixation while X represents the observational N fixation.

[Figure]

[Figure]

**Figure 4.** Simulated spatial distribution of  BNF rates (kg N ha$^{-1}$ yr$^{-1}$) in natural terrestrial ecosystems during 1990-2000 by considering the BNF effects.

(a)

[Figure]

(b)

[Figure]

(c)

[Figure]

(d)

[Figure]

[Figure]

**Figure 5.** Comparison between simulated and observed N$_2$O emissions (g N m$^{-2}$ day$^{-1}$) with two versions of the model (Red indicates the simulations considering BNF and blue indicates the simulations without considering BNF effects on N$_2$O emissions) at sites: (a) Temperate Forest (50.5°N, 8.5°E); (b) Tropical Forest (10°S, 63°W); (c) Savanna (10°N, 85°W); (d) Tropical Forest (6°S, 76°W).

---

## Editor Decision (ED1)

AWK.

can you give a description of the problem here? ITS laid out the is not abstract but their into

z

z s

s s s s

2          2

z

z

z

z

revise to " the state of Idaho in the US"

2      S

AWiz    2

2
2
2
2
the
2.

(BNF)

N₂          S

z

z

on $N_2$

z

FIX →

define abbreviation in Table legend

$N_2$

$N_2$

$N_2$

$N_2$

$N_2$

$N_2$

$N^2$

$N_2$

---

## Author Response (AR2)

Response letter

…However, the paper does have some grammatical issues and would benefit from having a professional editor to smooth over the language by improving sentence structure and grammatical flow of the manuscript.

*Response: Thank you for your suggestion. We have read over the manuscript and corrected these mistakes.*

One place that I see a need for improvement is at the end of the introduction-- I suggest adding a statement there about why the work needs to be done. This is stated in the abstract but not in the introduction.

*Response: We added a paragraph at the end of the introduction, from line 66 to line 72.*

Further, when referring to nitrogen fixation please use "N2" instead of N-- I marked all of the places this needs to be fixed.

*Response: Corrected.*

[revised manuscript text omitted]

(a)

[Figure]

(b)

[Figure]

[Figure]

[Figure]

(c)

[Figure]

[Figure]

**Figure 2.** Comparison between Modeled and Observed Nitrogen Fixation Rate at site level: (a) All sites, (b) Temperate Forest, (c) Tropical Forest (data listed in Table 2). Y is simulated $N_2$ fixation while X represents the observational $N_2$ fixation.

[Figure]

**Figure 3.** Simulated spatial distribution of BNF rates (kg $N_2$ ha$^{-1}$ yr$^{-1}$) in natural terrestrial ecosystems during 1990-2000 by considering the BNF effects.